# Hepatic and Extrahepatic Colorectal Metastases Have Discordant Responses to Systemic Therapy. Pathology Data from Patients Undergoing Simultaneous Resection of Multiple Tumor Sites

**DOI:** 10.3390/cancers13030464

**Published:** 2021-01-26

**Authors:** Luca Vigano, Pio Corleone, Shadya Sara Darwish, Nicolò Turri, Simone Famularo, Lorenzo Viggiani, Lorenza Rimassa, Daniele Del Fabbro, Luca Di Tommaso, Guido Torzilli

**Affiliations:** 1Division of Hepatobiliary and General Surgery—Department of Surgery, Humanitas Clinical and Research Center—IRCCS, 20089 Rozzano, Milan, Italy; luca.vigano@hunimed.eu (L.V.); pio.corleone@humanitas.it (P.C.); shadya@live.it (S.S.D.); simone.famularo@humanitas.it (S.F.); daniele.del_fabbro@humanitas.it (D.D.F.); 2Department of Biomedical Science, Humanitas University, 20090 Pieve Emanuele, Milan, Italy; n.turri2@studenti.uninsubria.it (N.T.); lorenzo.viggianidavalos@usz.ch (L.V.); Lorenza.rimassa@hunimed.eu (L.R.); luca.di_tommaso@hunimed.eu (L.D.T.); 3Department of General Surgery, Cattinara Hospital, University of Trieste, 34149 Trieste, Italy; 4Medical Oncology and Hematology Unit, Humanitas Clinical and Research Center—IRCCS, 20089 Rozzano, Milan, Italy; 5Department of Pathology, Humanitas Clinical and Research Center—IRCCS, 20089 Rozzano, Milan, Italy

**Keywords:** colorectal hepatic and extrahepatic metastases, systemic therapy, chemotherapy, targeted therapies, lung metastases, peritoneal metastases, lymph node metastases, response to chemotherapy, tumor regression grade

## Abstract

**Simple Summary:**

The standard treatment of patients with hepatic and extrahepatic metastases from colorectal cancer is systemic chemotherapy. We assume that this therapy has the same effectiveness on all disease foci, independent of the involved organ. The effectiveness of chemotherapy is assessed by the pathological response rate: the higher the response rate, the higher the effectiveness of chemotherapy. In the present manuscript, we analyzed patients undergoing resection of hepatic and extrahepatic metastases from colorectal cancer after preoperative chemotherapy. We observed unexpected heterogeneity of the response to chemotherapy of distant metastases from colorectal cancer according to the involved organ. Peritoneal metastases had the highest pathological response rate, which was much higher than the hepatic metastases, while lung and lymph node metastases had extremely poor response rates. Such inhomogeneous effectiveness of systemic treatment in different organs open new perspectives in the treatment of colorectal cancer with distant metastases and oncological research.

**Abstract:**

Background: Systemic therapy is the standard treatment for patients with hepatic and extrahepatic colorectal metastases. It is assumed to have the same effectiveness on all disease foci, independent of the involved organ. The present study aims to compare the response rates of hepatic and extrahepatic metastases to systemic therapy. Methods: All consecutive patients undergoing simultaneous resection of hepatic and extrahepatic metastases from colorectal cancer after oxaliplatin- and/or irinotecan-based preoperative chemotherapy were analyzed. All specimens were reviewed. Pathological response to chemotherapy was classified according to tumor regression grade (TRG). Results: We analyzed 45 patients undergoing resection of 134 hepatic and 72 extrahepatic metastases. Lung and lymph node metastases had lower response rates to chemotherapy than liver metastases (TRG 4–5 95% and 100% vs. 67%, *p* = 0.008, and *p* = 0.006). Peritoneal metastases had a higher pathological response rate than liver metastases (TRG 1–3 66% vs. 33%, *p* < 0.001) and non-hepatic non-peritoneal metastases (3%, *p* < 0.001). Metastases site was an independent predictor of pathological response to systemic therapy. Conclusions: Response to chemotherapy of distant metastases from colorectal cancer varies in different organs. Systemic treatment is highly effective for peritoneal metastases, more so than liver metastases, while it has a very poor impact on lung and lymph node metastases.

## 1. Introduction

Systemic therapy has a major impact on the prognosis of patients with stage IV colorectal cancer [1,2,3]. In patients with unresectable liver metastases, chemotherapy associated with antiangiogenic (anti-VEGF) or anti-epidermal growth factor receptor (anti-EGFR) monoclonal antibodies obtains a median survival even exceeding 30 months, allowing a secondary resection in some cases with major tumor shrinkage [3,4,5,6]. In resectable patients, systemic therapy prolongs progression-free survival after surgery and may lead to five-year survival rates approaching 50% [7,8,9,10]. Patients with both hepatic and extrahepatic metastases have lower survival expectancy than patients with liver-only disease, but systemic therapy improves their prognosis as well [1]. Selected patients, mainly those with pulmonary metastases, with adequate disease control by systemic therapies can be considered for resection of both hepatic and extrahepatic metastases, achieving adequate survival [11,12,13,14].

The response of liver metastases to chemotherapy is one of the strongest prognostic factors in patients undergoing surgery [15,16,17,18,19,20]. Both radiological and pathological evaluations have been associated with prognosis, even if some discrepancies between the two have been reported, with some patients with evident shrinkage at imaging having poor tumor regression at pathological examination [16,17]. The response to chemotherapy of extrahepatic metastases has been poorly investigated. Chemotherapy is assumed to have the same impact on all disease foci, independent of the involved organ. However, any evidence of the modulated effectiveness of systemic therapy on different organs could open new perspectives in treatment schedules, choice of treatment modality, and oncological research.

The present study aimed to compare the response of hepatic and extrahepatic colorectal metastases to systemic therapy. To achieve this aim, we analyzed a consecutive series of patients with simultaneous resection of hepatic and extrahepatic disease after systemic treatment.

## 2. Results

During the study period (2007–2018), 844 patients underwent liver resection for colorectal metastases. Of those, 68 (8%) had evidence of extrahepatic disease. Twenty-three patients were excluded for the following reasons: eight patients with hepatic and pulmonary metastases underwent a staged resection of the two organs; seven patients had hepatic disease progression while on preoperative chemotherapy (early years of the series); four patients did not receive preoperative chemotherapy; two patients had a complete response to chemotherapy and were operated upon for disease reappearance without further chemotherapy; two patients had no confirmation of extrahepatic disease at final pathology. Forty-five patients undergoing simultaneous resection of hepatic and extrahepatic metastases after effective preoperative chemotherapy were analyzed.

The most common sites of extrahepatic disease were the peritoneum (*n* = 21 patients), lung (*n* = 15), and lymph nodes (*n* = 14). Peritoneal metastatic involvement was always regional (local clearance). Five patients had extrahepatic metastases in multiple organs. Overall, 134 liver metastases and 72 extrahepatic metastases were resected. The median number of liver metastases per patient was two (range, 1–10), and the median number of extrahepatic metastases was one (range, 1–5). All patients underwent preoperative chemotherapy, including oxaliplatin and/or irinotecan, with a median number of cycles of seven (range, 4–24). Targeted therapies were associated with chemotherapy in 27 (60%) patients, being anti-VEGF the most common treatment (*n* = 19). Patients’ characteristics are summarized in Table 1.

### 2.1. Radiological Response

The radiological response was evaluable in 25 patients (15 patients had intraoperative detection of extrahepatic disease (peritoneal metastases in 14 and lymph node metastases in one), and five patients did not have imaging available for review). In the 25 analyzed patients, the extrahepatic disease had a complete response in one (4%, peritoneal metastasis), a partial response in 10 (40%), stable disease in 13 (52%), and disease progression in one (4%, pulmonary metastasis). In the same group, liver metastases had stable disease in six (24%) patients and a partial response in 19 (76%). Response to chemotherapy was less common in lung metastases than in liver metastases (in the whole series, 3/13 patients, 23% vs. 19/25, 76% *p* = 0.005; considering only the patients with hepatic and pulmonary metastases, 3/13, 23% vs. 9/13, 69% *p* = 0.049). Of note, one patient had a partial response of liver metastases but had progression of pulmonary metastasis. Table 2 summarizes the details of the radiological response analysis.

### 2.2. Pathological Response

#### 2.2.1. Per-Patient Analysis

In the per-patient analysis, liver metastases had a major pathological response (tumor regression grade 1–2, TRG 1–2) in two (5%) patients, a minor response (TRG 3) in six (13%), and no response (TRG 4–5) in 37 (82%). The extrahepatic disease had similar results: a major response in five (11%) patients, a minor response in five (11%), and no response in 35 (78%). Patients with pulmonary, lymph nodal, or adrenal metastases had no response to chemotherapy (TRG 4–5, 100% vs. 82% of liver metastases, *p* = 0.019). Conversely, 13 out of 21 (62%) patients with peritoneal metastases had a response to chemotherapy (TRG 1–3, vs. 18% of liver metastases, *p* < 0.001), including four (19%) patients with a major response.

#### 2.2.2. Per-Lesion Analysis

In the per-lesion analysis, 14 out of 134 (11%) liver metastases had a major response, 30 (22%) had a minor response, and 90 (67%) had no response. Overall, extrahepatic metastases had similar results: seven out of 72 (10%) had a major response, 17 (24%) had a minor response, and 48 (67%) had no response. However, lung and lymph node metastases had TRG 4–5 (no response to therapy) more often than liver metastases: 20 out of 21 (95%) lung metastases and all 15 lymph node metastases vs. 90 out of 134 (67%) liver metastases (*p* = 0.008 and *p* = 0.006, respectively). In contrast, peritoneal metastases had a pathological response to chemotherapy (TRG 1–3) more often than the other disease sites: 23 out of 35 (66%) peritoneal metastases vs. 44 out of 134 (33%) liver metastases (*p* < 0.001) and one out of 38 (3%) non-peritoneal non-hepatic metastases (*p* < 0.001). One-fifth of patients with peritoneal metastases (*n* = 7) had a major pathological response to chemotherapy. Table 3 and Figure 1 summarize the details of the pathological response analysis.

At the multivariate analysis (Table 4), the metastasis site was an independent predictor of pathological response to chemotherapy: peritoneal metastases were a positive predictor of TRG 1–3 (odds ratio (OR) = 12.709, *p* < 0.001), while lung metastases were a negative one (OR = 0.057, *p* = 0.014). Lymph nodal involvement was omitted from multivariate analysis because it perfectly predicted failure (TRG 4–5 in all patients). Additional predictors of pathological tumor response (TRG 1–3) were the administration of anti-VEGF or anti-EGFR therapies (OR = 9.748, *p* = 0.001, and OR = 69.830, *p* < 0.001, respectively) and metastasis size (OR = 0.961, *p* = 0.049). The chemotherapy regimen (oxaliplatin or irinotecan), KRAS mutational status, and primary tumor site were not associated with pathological tumor response.

## 3. Discussion

In patients with stage IV colorectal cancer, we assume that systemic therapy is equally effective for all tumor sites. This assumption is mainly based on radiological tumor response to treatment. Even if the latter parameter is strictly associated with prognosis [18,19], it has some limitations. First, a discrepancy between radiological and pathological response to chemotherapy may occur [16,17]. In the present study, liver metastases had a radiological response to chemotherapy in two-thirds of patients but a pathological response in only one-third. Second, the small size of extrahepatic metastases, notably of pulmonary and peritoneal metastases, can preclude the reliable application of the RECIST criteria. We observed a lower radiological response rate of lung metastases in comparison with hepatic metastases, but the first group had a small size (median 7 mm) that could bias the results. Finally, most peritoneal nodules are undetectable at imaging (in the present series, 14 out of 21 patients). Thus, far, the evaluation of pathological response to therapy is mandatory to make any conclusions about the treatment effectiveness on metastases in different organs.

Some studies highlighted an inhomogeneous pathological response among different metastases in the same patient, but most data concern discrepancies among liver metastases [15,16,21,22]. This phenomenon could be explained by inter-metastases genetic heterogeneity [23]. A single study analyzed pathological tumor response in different organs. Gervaz et al. compared TRG values of primary colorectal cancer and liver metastases in patients undergoing simultaneous resection of both disease sites [24]. They reported lower response rates of primary tumors and their loco-regional lymph node metastases in comparison with hepatic nodules. In the present series, we had the opportunity to analyze a consecutive series of 45 patients undergoing simultaneous resection of hepatic and extrahepatic metastases. Lung metastases are usually resected by staged procedures [13,25,26], i.e., lung resection at least one month after liver resection, precluding any reliable comparative analysis between the two organs. Our approach by a thoracoabdominal incision allowed simultaneous resection of hepatic and pulmonary metastases in a non-negligible number of cases (21 nodules in 15 patients) [27].

We observed heterogeneous pathological responses among metastases in different organs of the same patient. Three main findings deserve consideration. First, lung metastases had an extremely poor pathological response: all nodules, but one had no regression (TRG 4–5), and the remaining nodule had a minor response (TRG 3). These data are much worse than those of liver metastases (TRG 1–3 in 30% of nodules, a major response in 10%). Such a result is even more surprising if we consider the size of lung metastases (median 7 mm) and that, according to our analysis, the smaller the metastasis, the higher the chance of pathological tumor response. The prognostic impact of colorectal lung metastases is still a matter of debate, especially in patients with limited pulmonary tumor burden. Even if resection improves prognosis, some researchers reported a survival benefit after liver-only resection leaving pulmonary metastases in place [13,28,29]. Preliminary series of liver transplantation for colorectal metastases reported a non-negligible proportion of patients having lung recurrence with a minimal impact on survival [30,31]. The present data, demonstrating the poor response of lung metastases to systemic chemotherapy, confirmed that pulmonary nodules require a separate evaluation, and their most appropriate management remains uncertain, even if some evidence in favor of perioperative chemotherapy in these patients has been reported [32,33]. Second, lymph node metastases had a poor pathological response (no TRG 1–3). Our data agree with that of Gervaz et al., who outlined a lower response of peri-colonic lymph nodes to chemotherapy than liver metastases [24]. In 2008, Adam et al. reported poor prognoses in patients with distant lymph node metastases even after disease control by chemotherapy [34]. The lack of pathological response to chemotherapy evident in our analysis could explain those data. Finally, peritoneal metastases had the most favorable response (tumor regression in two-thirds of cases, a major response in one-fifth). Even if peritoneal carcinomatosis is a poor prognostic index, its responsiveness to chemotherapy has been documented [35,36]. A debate is ongoing regarding systemic and intraperitoneal chemotherapy [36,37], with the data from the Prodige 7 trial suggesting similar results between the two [38].

The major contribution of targeted therapies to the treatment of colorectal metastases was confirmed [2,3,4,5,6]. The association of both anti-VEGF and anti-EGFR monoclonal antibodies to chemotherapy increased the pathological tumor response rate. However, our results could provide new perspectives in the treatment of stage IV colorectal cancer and oncological research. The heterogeneous effectiveness of systemic therapy in different organs should be considered, and any treatment should be analyzed for its specific effectiveness on different metastatic sites. In lung metastases, given their attitude to progress slowly and poor response to chemotherapy, loco-regional treatments could gain interest, especially if featured by low risk and invasiveness, such as radiotherapy [14,39]. Conversely, the high effectiveness of systemic treatment on peritoneal metastases could further justify an aggressive local approach. Finally, the heterogeneity of the response should be investigated at the anatomical, immunological, and genetic/epigenetic levels to understand its mechanisms and to increase the effectiveness of treatments.

There are some limitations of the present analysis. First, this is a retrospective study that included a limited number of patients and extrahepatic metastases. We selected only patients that underwent simultaneous resection of hepatic and extrahepatic metastases to compare the response to chemotherapy of disease foci in different organs. Such patients are the minority of candidates to surgery and usually have a limited extrahepatic tumor burden: we had to consider 844 patients undergoing liver resection across 12 years to identify 45 patients with 72 extrahepatic metastases. Second, despite the excellent radiological response rate, the observed pathological response rate was quite low and lower than expected. State-of-the-art chemotherapy was administered, and standard timing chemotherapy-surgery was respected. The aggressiveness of the disease (hepatic and extrahepatic) and the high proportion of KRAS mutated tumors could justify these data. Regardless of this, the reliability of the present analysis is robust because the pathological data of different organs of the same patient undergoing simultaneous resection were compared. We used standard pathological scores [15]. Third, the small size of extrahepatic metastases could put in doubt the reliable evaluation of pathological response. However, both lung and peritoneal metastases had a small median size (7 vs. 9 mm, respectively) but had a large difference in TRG values. Fourth, the impact of gene mutations on TRG was not fully explored: the present series included 20 patients with KRAS mutation, but none with NRAS or BRAF mutation. Finally, only responders to chemotherapy were selected (potential overestimation of the effectiveness of systemic treatment), and patients with a complete response to systemic treatment were excluded (underestimation). Large prospective observational studies including patients independently of their radiological response to chemotherapy are needed to validate the present data and further analyze the association between TRG and genetic mutational status.

## 4. Materials and Methods

All consecutive patients undergoing liver resection for colorectal metastases between January 2007 and December 2018 were retrospectively analyzed. The following inclusion criteria were used: (1) patients with hepatic and extrahepatic metastases from colorectal cancer undergoing complete resection; (2) simultaneous resection of hepatic and extrahepatic disease; and (3) ≥2 months preoperative oxaliplatin- and/or irinotecan-based chemotherapy. Patients undergoing a staged resection of hepatic and extrahepatic metastases in two separate operations were excluded.

All specimens of both hepatic and extrahepatic metastases were reviewed by an expert hepatobiliary pathologist (LDT). The pathological response was classified according to the TRG [15]. We applied the same classification used for liver metastases to extrahepatic metastases. TRG values were then grouped into the major response (TRG 1–2), minor response (TRG 3), and no response (TRG 4–5) categories [15]. Both per-patient and per-lesion analyses were performed. In the per-patient analysis, if multiple metastases in the same organ had discordant TRG values, the highest value (the poorest response) was considered. All the available imaging modalities before and after preoperative chemotherapy were reviewed. Radiological response to chemotherapy of hepatic and extrahepatic metastases was classified according to RECIST criteria version 1.1 [40].

Our management of patients with liver metastases from colorectal cancer has been previously reported [41,42]. A multidisciplinary expert team established the therapeutic strategy for all patients. Surgery was considered only in patients exhibiting disease control by chemotherapy (stable disease or partial response) amenable to complete surgery of both hepatic and extrahepatic metastases. Surgery was scheduled four to six weeks after the end of chemotherapy (six weeks in patients receiving anti-VEGF targeted therapies). Selected patients with lung metastases had simultaneous hepatic and pulmonary resection through a thoracoabdominal approach [27].

The present study is a retrospective analysis of a prospectively maintained database. Categorical variables were compared using the chi-squared test or Fisher’s exact test, as appropriate. One continuous variable (metastases size) was compared between groups using Mann–Whitney *U*-test. A multivariable logistic regression model was performed to identify independent predictors of pathological response to chemotherapy (TRG 1–3). A *p*-value of <0.05 was considered significant for all tests. Stata 15 software package was used for all the analyses.

## 5. Conclusions

The present study revealed unexpected heterogeneity of the response to systemic therapy of distant metastases from colorectal cancer according to the involved organ. Peritoneal metastases had the highest pathological response rate, which was much higher than the liver, while lung and lymph node metastases had extremely poor response rates. Such inhomogeneous effectiveness of systemic treatment in different organs could reveal new perspectives in treatment strategies and research.

## Figures and Tables

**Figure 1 cancers-13-00464-f001:**
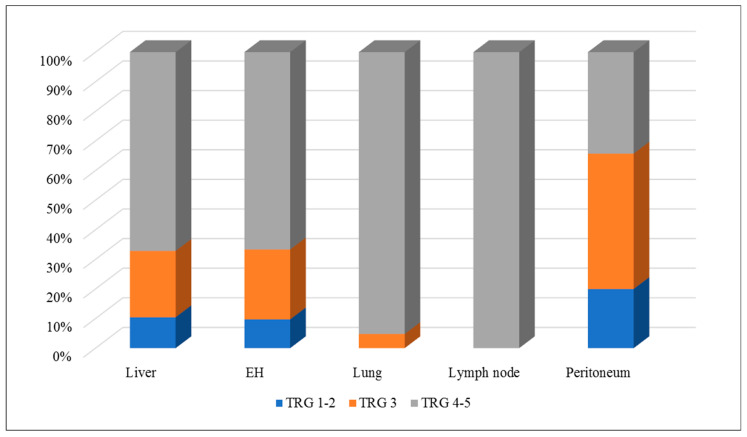
Per-lesion analysis of the pathological response to chemotherapy according to the metastasis site.

**Table 1 cancers-13-00464-t001:** Patients’ characteristics and chemotherapy details.

Demographical Characteristics	*n* = 45
Age, years, median (range)	59 (34–76)
Sex (M:F)	27 (60%):18 (40%)
Primary tumor site (right colon: left colon: rectum)	15 (33%):17 (38%):13 (29%)
Microsatellite instability	0
Synchronous liver metastases	31 (69%)
Number of liver metastases, median (range)	2 (1–10)
Size of liver metastases, mm, median (range)	14 (1–180)
KRAS mutated status (status available in 36 patients)	20/36 (56%)
NRAS mutated status (status available in 27 patients)	0
BRAF mutated status (status available in 26 patients)	0
Extrahepatic disease		
Lung	15
Number of nodules, median (range)	1 (1–3)
Size, mm, median (range)	7 (3–37)
Lymph nodes	14
Number of nodules, median (range)	1 (1–2)
Size, mm, median (range)	15 (10–40)
Peritoneum	21
Number of nodules, median (range)	1 (1–5)
Size, mm, median (range)	9 (2–40)
Adrenal gland	1
Chemotherapy details		
Regimen	
Oxaliplatin	22 (49%)
Irinotecan	21 (47%)
Oxaliplatin + Irinotecan	2 (4%)
Targeted therapies	27 (60%)
Anti-VEGF	19 (42%)
Anti-EGFR	8 (18%)
Number of cycles, median (range)	7 (4–24)
Number of lines, median (range)	1 (1–2)

**Table 2 cancers-13-00464-t002:** The radiological response of hepatic and extrahepatic metastases from colorectal cancer to systemic therapy.

Per-Patient Analysis (*n* = 25)					
	Liver *n* = 25	Extrahepatic *n* = 25	Lung *n* = 13	Lymph Node *n* = 9	Peritoneum *n* = 7
Complete response	-	1 (4%)	-	-	1 (14%)
Partial response	19 (76%)	10 (40%)	3 (23%)	5 (56%)	2 (29%)
Stable disease	6 (24%)	13 (52%)	9 (69%)	4 (44%)	4 (57%)
Disease progression	-	1 (4%)	1 (8%)	-	-

**Table 3 cancers-13-00464-t003:** The pathological response of hepatic and extrahepatic metastases from colorectal cancer to systemic therapy.

Per-patient Analysis (*n* = 45)					
TRG	Liver *n* = 45	Extrahepatic *n* = 45	Lung *n* = 15	Lymph Node *n* = 14	Peritoneum *n* = 21
TRG 1–2	2 (5%)	5 (11%)	-	-	4 (19%)
TRG 3	6 (13%)	5 (11%)	-	-	9 (43%)
TRG 4–5	37 (82%)	35 (78%)	15 (100%)	14 (100%)	8 (38%)
**Per-Lesion Analysis (*n* = 134 Liver Metastases and *n* = 72 Extrahepatic Metastases)**					
**TRG**	**Liver** ***n* = 134**	**Extrahepatic** ***n* = 72**	**Lung** ***n* = 21**	**Lymph node** ***n* = 15**	**Peritoneum** ***n* = 35**
TRG 1–2	14 (11%)	7 (10%)	-	-	7 (20%)
TRG 3	30 (22%)	17 (24%)	1 (5%)	-	16 (46%)
TRG 4–5	90 (67%)	48 (67%)	20 (95%)	15 (100%)	12 (34%)

**Table 4 cancers-13-00464-t004:** Univariate and multivariate analyses of the predictors of pathological tumor response to systemic therapy (TRG1–3).

Parameter		TRG 1–3	Univariate	Multivariate Analysis		
				*p*	OR	95% CI
Metastasis site	Liver	44 (32.8%)	<0.001		1	
	Lung	1 (4.8%)		0.014	0.057	0.006–0.566
	Lymph node	- (0%)		Omitted (perfect prediction of failure)		
	Peritoneum	23 (65.7%)		<0.001	12.709	3.102–52.063
	Adrenal	- (0%)		Omitted (perfect prediction of failure)		
Metastasis size, mm		13.9 ± 12.9 (vs. 20.4 ± 26.3)	0.023	0.049	0.961	0.923–0.999
Primary tumor site	Right colon	28 (43.1%)	0.018		1	
	Left colon	32 (33.7%)		0.635	0.774	0.270–2.223
	Rectum	8 (17.4%)		0.196	0.364	0.078–1.687
KRAS status *	Wild type	19 (29.7%)	0.101	0.218	0.385	0.084–1.757
	Mutated	40 (42.6%)				
Oxaliplatin	Y	25 (28.7%)	0.265	0.178	8.533	0.376–193.728
	*n*	43 (36.1%)				
Irinotecan	Y	44 (36.4%)	0.222	0.151	9.905	0.432–226.851
	N	24 (28.2%)				
Anti-VEGF	Y	33 (39.3%)	0.112	0.001	9.748	2.498–38.041
	N	35 (28.7%)				
Anti-EGFR	Y	23 (56.1%)	<0.001	<0.001	69.830	8.977–543.186
	N	45 (27.3%)				
Number of lines	1	61 (33.0%)	0.973	0.402	1.927	0.416–8.922
	>1	7 (33.3%)				

* KRAS status was available in 36 patients, 158 metastases. Metastases size is expressed as mean ± SD. OR, odds ratio; 95% CI, 95% confidence intervals.

## Data Availability

The data presented in this study are available on request from the corresponding author. The data are not publicly available due to privacy restrictions.

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
