# Peer review of "Hepatic and Extrahepatic Colorectal Metastases Have Discordant Responses to Systemic Therapy. Pathology Data from Patients Undergoing Simultaneous Resection of Multiple Tumor Sites"

_cancers, 2021, doi:10.3390/cancers13030464_

Round 1

Reviewer 1 Report

In the present manuscript, the authors compare the response of hepatic and extrahepatic colorectal metastases to systemic therapy. From 2007-2018, a consecutive series of 45 patients with simultaneous resection of hepatic  and extrahepatic metastases after systemic treatment was examined. Extrahepatic metastatic sites were peritoneum in 21 patients with local tumor burden and complete surgical clearance, lung in 15 patients, lymph nodes in 14 patients. In total, 134 liver metastases and 72 extrahepatic metastases were resected.

Radiologic response to therapy was only evaluable in 25 patients (15 intraoperatively detected extrahepatic tumor burden, five with incomplete imaging studies. In these few patients, liver metastases showed significantly more response to preoperative chemotherapy (76% - 19 patients) than extrahepatic metastases (44% - 11 patients)

In contrast, pathological response was seen in only 18% of patients, comparable with 22% pathological response in extrahepatic metastases. However, within the extrahepatic metastases groupd. Pulmonary, lymph node or adrenal metastases showed no response to chemotherapy, but peritoneal metastases had response in 62% of patients.

At multivariate analysis, the metastatic site was an independent predictor of pathological response to chemotherapy, with peritoneal metastases as positive predictor of tumor response, lung metastases as negative predictor. Additional predictors of pathological response to systemic therapy in multivariate alanysis were administration of antibody therapy and size of metastases.

As a conclusion, the authors state that radiological response is not an adequate measure of tumor response to therapy and that systemic therapy is not equally effective in different metastatic sites. Especially local peritoneal metastases, which often are not adequately presented in radiological imaging, show a good response to preoperative chemotherapy. Therefore, pathological response is mandatory to adequately assess efficacy of systemic therapy.

Overall, the study seems to be well conducted and reports interesting and relevant results. The data are well presented. I have no major concerns. Minor concerns (which are addressed in the discussion by the authors themselves) are:

-in total, only a small number of patients was evaluated

- only responders to chemotherapy were evaluated

- only few extrahepatic metastases were included

- minor typos, e.g. abstract Line 47/48: incomplete sentence

Author Response

We thank the reviewer for his/her comments, and the very careful evaluation of our study. We think that the manuscript has been considerably improved and we hope that he/she will consider our replies adequate.

Reviewer #1

In the present manuscript, the authors compare the response of hepatic and extrahepatic colorectal metastases to systemic therapy. From 2007-2018, a consecutive series of 45 patients with simultaneous resection of hepatic  and extrahepatic metastases after systemic treatment was examined. Extrahepatic metastatic sites were peritoneum in 21 patients with local tumor burden and complete surgical clearance, lung in 15 patients, lymph nodes in 14 patients. In total, 134 liver metastases and 72 extrahepatic metastases were resected.

Radiologic response to therapy was only evaluable in 25 patients (15 intraoperatively detected extrahepatic tumor burden, five with incomplete imaging studies. In these few patients, liver metastases showed significantly more response to preoperative chemotherapy (76% - 19 patients) than extrahepatic metastases (44% - 11 patients)

In contrast, pathological response was seen in only 18% of patients, comparable with 22% pathological response in extrahepatic metastases. However, within the extrahepatic metastases groupd. Pulmonary, lymph node or adrenal metastases showed no response to chemotherapy, but peritoneal metastases had response in 62% of patients.

At multivariate analysis, the metastatic site was an independent predictor of pathological response to chemotherapy, with peritoneal metastases as positive predictor of tumor response, lung metastases as negative predictor. Additional predictors of pathological response to systemic therapy in multivariate alanysis were administration of antibody therapy and size of metastases.

As a conclusion, the authors state that radiological response is not an adequate measure of tumor response to therapy and that systemic therapy is not equally effective in different metastatic sites. Especially local peritoneal metastases, which often are not adequately presented in radiological imaging, show a good response to preoperative chemotherapy. Therefore, pathological response is mandatory to adequately assess efficacy of systemic therapy.

 Overall, the study seems to be well conducted and reports interesting and relevant results. The data are well presented. I have no major concerns. Minor concerns (which are addressed in the discussion by the authors themselves) are:

- in total, only a small number of patients was evaluated

- only responders to chemotherapy were evaluated

- only few extrahepatic metastases were included

We thank the reviewer for his/her comment. We agree with the study limitations that have been mentioned. We have better detailed the limitations as follows:

“There are some limitations of the present analysis. First, this is a retrospective study that included a limited number of patients and extrahepatic metastases. We selected only patients that underwent simultaneous resection of hepatic and extrahepatic metastases to compare the response to chemotherapy of disease foci in different organs. Such patients are the minority of candidates to surgery and usually have a limited extrahepatic tumor burden: we had to consider 844 patients undergoing liver resection across 12 years to identify 45 patients with 72 extrahepatic metastases. Second, despite the excellent radiological response rate, the observed pathological response rate was quite low, and lower than expected. State-of-the-art chemotherapy was administered, and standard timing chemotherapy-surgery was respected. The aggressiveness of the disease (hepatic and extrahepatic) and the high proportion of KRAS mutated tumors could justify these data. Regardless of this, the reliability of the present analysis is robust because the pathological data of different organs of the same patient undergoing simultaneous resection were compared. We used standard pathological scores [15]. Third, the small size of extrahepatic metastases could put in doubt the reliable evaluation of pathological response. However, both lung and peritoneal metastases had a small median size (7 vs. 9 mm, respectively), but had a large difference in TRG values. Fourth, the impact of gene mutations on TRG was not fully explored: the present series included 20 patients with KRAS mutation, but none with NRAS or BRAF mutation. Finally, only responders to chemotherapy were selected (potential overestimation of the effectiveness of systemic treatment), and patients with a complete response to systemic treatment were excluded (underestimation). Large prospective observational studies including patients independently of their radiological response to chemotherapy are needed to validate the present data and further analyze the association between TRG and genetic mutational status.”

- minor typos, e.g. abstract Line 47/48: incomplete sentence

We have checked the manuscript for typing errors.

Reviewer 2 Report

The paper is of interest, I have some points:

  • The authors reported discrepant TRG values between lung and hepatic metastases. This is quite surprising considering that only a minority of patients has a discordant radiological response to chemotherapy. Please comment. Could it be related to the analyzed cohort?
  • Some papers reported series of resected extrahepatic mets from primary colorectal cancer. For example this paper (Cancer Med. 2016 Feb;5(2):256-64) stressed the importance of adjuvant treatment after lung metastasectomy (with significant improvement of OS and PFS). Moreover, KRAS mutation was reported as a negative prognostic factor (and you reported it as well). You evaluated response to preop chemotherapy and TRG. What about patients who received postop chemotherapy? Do you have data on them? A comparison between the two series may be of interest.
  • Most peritoneal metastases were not visible at preoperative imaging. How can the authors reliably evaluate their response to chemotherapy?
  • The authors should further comment on the association between mutational status and TRG and how it could have impacted their result

Author Response

We thank the reviewer for his/her comments, and the very careful evaluation of our study. We think that the manuscript has been considerably improved and we hope that he/she will consider our replies adequate.

Reviewer #2

The paper is of interest, I have some points:

  • The authors reported discrepant TRG values between lung and hepatic metastases. This is quite surprising considering that only a minority of patients has a discordant radiological response to chemotherapy. Please comment. Could it be related to the analyzed cohort?

We thank the reviewer for his/her comment. Indeed, we usually observe a concordant radiological response to chemotherapy of metastases in the liver and the lung, but the reviewer should consider that radiological and pathological response can be discrepant. As recently reported for liver metastases (Viganò L. et al. Ann Surg. 2013, Nov;258(5):731-40; Brouquet A. et al. Ann Surg Oncol. 2020, Aug;27(8):2877-2885), up to one-third of patients with a radiological response to chemotherapy has no evidence of response at pathological analysis (TRG 4-5). We can assume that a similar phenomenon occurs for lung metastases. A large cohort of patients is needed to confirm present data, but we consider our data robust because they rely on pathological data analyzed in patients undergoing simultaneous resection of hepatic and extrahepatic disease.

A further potential limitation of the present study should be considered. Only responders to chemotherapy were selected (potential overestimation of the effectiveness of systemic treatment), and patients with a complete response to systemic treatment were excluded (underestimation).

We detailed the potential discrepancy between radiological and pathological response to chemotherapy as follows:

“Both radiological and pathological evaluations have been associated with prognosis, even if some discrepancies between the two have been reported, with some patients with evident shrinkage at imaging having poor tumor regression at pathological examination [16,17].”

“In patients with stage IV colorectal cancer, we assume that systemic therapy is equally effective for all tumor sites. This assumption is mainly based on radiological tumor response to treatment. Even if the latter parameter is strictly associated with prognosis [18,19], it has some limitations. First, a discrepancy between radiological and pathological response to chemotherapy may occur [16,17].”

We better detailed the limitations of the study in the discussion as follows:

“Finally, only responders to chemotherapy were selected (potential overestimation of the effectiveness of systemic treatment), and patients with a complete response to systemic treatment were excluded (underestimation). Large prospective observational studies including patients independently of their radiological response to chemotherapy are needed to validate the present data and further analyze the association between TRG and genetic mutational status.”

  • Some papers reported series of resected extrahepatic mets from primary colorectal cancer. For example this paper (Cancer Med. 2016 Feb;5(2):256-64) stressed the importance of adjuvant treatment after lung metastasectomy (with significant improvement of OS and PFS). Moreover, KRAS mutation was reported as a negative prognostic factor (and you reported it as well). You evaluated response to preop chemotherapy and TRG. What about patients who received postop chemotherapy? Do you have data on them? A comparison between the two series may be of interest.

We agree with the reviewer that adjuvant chemotherapy may impact patients’ prognosis. In the present series, 27 out of 45 patients had adjuvant chemotherapy after resection. However, we did not analyze the long-term outcome of patients. We focused on the effectiveness of preoperative chemotherapy and radiological and pathological response rates. So far, we cannot investigate the association between present data and adjuvant chemotherapy. We believe that the analysis of prognostic factors in such a small and heterogeneous cohort of patients would be scarcely informative. Nevertheless, according to the reviewer’s suggestion, we mentioned the potential prognostic impact of adjuvant chemotherapy in the discussion as follows:

“The present data, demonstrating the poor response of lung metastases to systemic chemotherapy, confirmed that pulmonary nodules require a separate evaluation, and their most appropriate management remains uncertain, even if some evidence in favor of peri-operative chemotherapy in these patients have been reported [32,33].”

  • Most peritoneal metastases were not visible at preoperative imaging. How can the authors reliably evaluate their response to chemotherapy?

We agree with the reviewer’s comment. Only 7 of the 21 patients with peritoneal metastases had the peritoneal nodules evident at pre-chemotherapy imaging. The remaining 14 patients had intraoperative detection of peritoneal metastases. Accordingly, we cannot reliably evaluate the radiological response to chemotherapy of peritoneal metastases. Data have been reported in Table 2. Further, this limitation was mentioned in the text as follows:

Results 

“Radiological response was evaluable in 25 patients (15 patients had intraoperative detection of extrahepatic disease (peritoneal metastases in 14 and lymph node metastases in one), and five patients did not have imaging available for review).”

Discussion

“In patients with stage IV colorectal cancer, we assume that systemic therapy is equally effective for all tumor sites. This assumption is mainly based on radiological tumor response to treatment. Even if the latter parameter is strictly associated with prognosis [18,19], it has some limitations. … Second, the small size of extrahepatic metastases, notably of pulmonary and peritoneal metastases, can preclude the reliable application of the RECIST criteria. We observed a lower radiological response rate of lung metastases in comparison with hepatic metastases, but the first group had a small size (median 7 mm) that could bias the results. Finally, most peritoneal nodules are undetectable at imaging (in the present series, 14 out of 21 patients). Thus far, the evaluation of pathological response to therapy is mandatory to make any conclusions about the treatment effectiveness on metastases in different organs. ”

Nevertheless, the reviewer should consider that the main results of the present study rely on the pathological data that are independent of the detectability of peritoneal metastases at pre-chemotherapy imaging.

This is mentioned in the discussion as follows:

“Regardless of this, the reliability of the present analysis is robust because the pathological data of different organs of the same patient undergoing simultaneous resection were compared.”

  • The authors should further comment on the association between mutational status and TRG and how it could have impacted their result

This is an important point because the mutational status of colorectal tumors is associated with prognosis. In the present analysis, the KRAS mutational status was available in 36 patients and was mutated in 20. KRAS mutational status was not associated with the TRG neither at univariate nor at multivariate analysis. A larger population is needed to reach conclusive data. Concerning the NRAS status and BRAF status, all patients with available data were wild type, precluding the possibility to explore any association between gene mutations and TRG.

We added data about NRAS and BRAF status in Table 1.

We added this limitation in the discussion as follows:

“There are some limitations of the present analysis. …. Fourth, the impact of gene mutations on TRG was not fully explored: the present series included 20 patients with KRAS mutation, but none with NRAS or BRAF mutation. Finally, only responders to chemotherapy were selected (potential overestimation of the effectiveness of systemic treatment), and patients with a complete response to systemic treatment were excluded (underestimation). Large prospective observational studies including patients independently of their radiological response to chemotherapy are needed to validate the present data and further analyze the association between TRG and genetic mutational status.”
